# Volatile Organic Compound Profiles Associated with Microbial Development in Feedlot Pellets Inoculated with *Bacillus amyloliquefaciens* H57 Probiotic

**DOI:** 10.3390/ani11113227

**Published:** 2021-11-11

**Authors:** Thi Thuy Ngo, Peter Dart, Matthew Callaghan, Athol Klieve, David McNeill

**Affiliations:** 1School of Veterinary Science, The University of Queensland, Gatton, QLD 4343, Australia; d.mcneill@uq.edu.au; 2Faculty of Animal Science, Vietnam National University of Agriculture, Hanoi 131000, Vietnam; 3School of Agriculture and Food Sciences, The University of Queensland, Gatton, QLD 4343, Australia; p.dart@uq.edu.au; 4Ridley AgriProducts Pty Ltd., Toowong, QLD 4066, Australia; Matthew.Callaghan@ridley.com.au; 5Queensland Alliance for Agriculture and Food Innovation, The University of Queensland, St. Lucia, QLD 4069, Australia; a.klieve@uq.edu.au

**Keywords:** probiotic, *Bacillus amyloliquefaciens* H57, volatile organic compounds, microbial volatile organic compounds, microbial development, feedlot pellets

## Abstract

**Simple Summary:**

Our study aimed to confirm that the probiotic, *Bacillus amyloliquefaciens* strain H57 (H57), manufactured into grain-rich stockfeed pellets, would help maintain the various types of odours of the pellets during an extended storage. Pellets treated with (H57) or without (control, C) were stored either at room temperature or at 5 °C over 3 months. The odours were identified in the pellets, stored at 0, 1, 2 and 3 months, by a gas analysis technique. The change of odour types was greatest in the C pellets stored for 3 months at room temperature (CA3) than all other pellet treatments. The odour types of the H57 pellets aged 2 or 3 months at room temperature were similar to that of C pellets aged 1 or 2 months. Nine odour types of microbial origin were related to the change observed in CA3. These odour types have been previously identified in grains spoiled by mould and thus deserve further evaluation as indicators of the types of mould against which H57 protects as a feed inoculum. These results suggest that H57 can help to maintain the odour of stockfeed pellets, by reducing the rate of microbial spoilage during storage.

**Abstract:**

Mould and bacterial contamination releases microbial volatile organic compounds (mVOCs), causing changes in the odour profile of a feed. *Bacillus amyloliquefaciens* strain H57 (H57) has the potential ability to inhibit microbial growth in animal feeds. This study tested the hypothesis that H57 influences the odour profile of stored feedlot pellets by impeding the production of mVOCs. The emission of volatile organic compounds (VOCs) of un-inoculated pellets and those inoculated with H57, stored either at ambient temperature (mean 22 °C) or at 5 °C, was monitored at 0, 1, 2, and 3 months by gas chromatography–mass spectrometry. Forty VOCs were identified in all the pellet samples analysed, 24 of which were potentially of microbial and 16 of non-microbial origin. A score plot of the principal component analysis (PCA) showed that the VOC profiles of the pellets stored at ambient temperature changed more rapidly over the 3 months than those stored at 5 °C, and that change was greater in the un-inoculated pellets when compared to the inoculated ones. The bi-plot and correlation loading plots of the PCA indicated that the separation of the un-inoculated pellets from the other treatments over the 3 months was primarily due to nine mVOCs. These mVOCs have been previously identified in grains spoiled by fungi, and could be considered potential markers of the types of fungi that H57 can protect pellets against. These data indicate the ability of H57 to maintain the odour profile and freshness of concentrated feed pellets. This protective influence can be detected as early as 3 months into ambient temperature storage.

## 1. Introduction

Volatile organic compounds are responsible for the characteristic odours of grass pasture [1,2,3,4] and concentrate feeds [5]. Kirstine [6] analysed the VOCs from white clover and pasture grass that primarily consisted of perennial ryegrass and found that both freshly cut clover and grass had high concentrations of oxygenated compounds: (Z)-3-hexen-1-ol, (Z)-2-hexen-1-ol, (E)-2-hexenal, (Z)-3-hexenal, and (Z)-3-hexenyl acetate. These oxygenated hydrocarbons are primarily responsible for the “green-leaf odour” that is characteristic of freshly harvested grass [7]. Likewise, Mayland [8] found that the “green-leaf odour” of both the fresh herbage and hay of tall fescue were due to the presence of four VOC alcohols (hexanol, (E)-2-hexenol, (E)-3-hexenol and (Z)-3-hexenol), as well as four VOC aldehydes (hexanal, (E)-2-hexenal, (E)-3-hexenal and (Z)-3-hexenal). Rapisarda [9] studied the odour characteristics of oat grains and found that high levels of VOC aldehydes, such as 2,4-nonadienal, (Z)-2-octenal, hexanal, and (E)-2-nonenal resulted in “nutty-like” and “green leaf-like” odours. In contrast, three sulfur-containing VOCs (methyl ethyl sulphide, thiophene, and dimethyl trisulfide) that were identified in dehydrated alfalfa, corn middlings, corn gluten meal, sunflower meal, and wheat brans created the “garlic” off-odours often related to these feeds. However, these studies did not discuss VOCs in relation to mould growth or bacterial contamination.

The growth of microorganisms within feeds produces a range of VOCs of microbial origin (mVOCs); some of which may have an adverse impact on feed quality by altering the odour profile of feeds [10,11]. Sinha [12] found that a high concentration of three mVOCs: 3-octanone, 1-octene-3-ol, and 3-methyl-l-butanol, in wheat grain stored in non-ventilated bins, was correlated with heavy infestations of *Alternaria alternata* and postharvest storage fungi (*Penicillium* spp. and *Aspergillus* spp.). Similarly, Jeleń and Wasowicz [13] found l-octen-3-ol, 3-methyl butanol, and 3-octanone in spoiled grains with a mouldy off-odour. Jeleń [14] also showed that the mVOCs geosmin and 2-methylisoborneol were responsible for the musty, earthy and mouldy off-odour of wheat grain that was contaminated by *Penicillium*, *Aspergillus flavus*, *Aspergillus glaucus,* as well as bacteria. *Aspergillus fumigatus*, *Aspergillus repens*, *Mucor* spp., *Absidia* spp., and *Emericella nidulans* have also been proposed as causes of off-odours and reduced quality in both alfalfa hay [15] and baled fescue forage [16]. The associated mVOC profiles produced by these fungi were not determined.

Microbial inoculants have the potential to inhibit microbial growth and, consequently, the development of off-odours in animal feeds. The common use of lactic-acid-producing anaerobes as biocontrol inoculants in order to preserve the quality of silage has been reviewed by Wittenberg [17] and Borreani [18]. Dulcet [19] found that the commercial inoculant *Bacillus pumilis* 1155 could reduce the increases in temperature caused by the microbial respiration within hay which was baled at different densities and moistures (20% and 28%). This result is similar to that reported by Brown and Dart [20] for H57, marketed as HayRite. Strains of *B. amyloliquefaciens* can produce mVOCs with antimicrobial activity [21]. Yuan [22] found that 11 mVOCs produced by *B. amyloliquefaciens* NJN-6 inhibited the growth of the plant-pathogenic fungus *Fusarium oxysporum* f. sp. *cubense*. Additionally, Raza [23,24] reported that *B. amyloliquefaciens* SQR-9 and *B. amyloliquefaciens* T-5 release mVOCs with antibacterial activity against *Ralstonia solanacearum*. H57 has been promoted for the biological control of spoilage moulds such as species of *Aspergillus*, *Cephalosprium*, *Eurotium*, *Fusarium*, *Penicillium,*
*Scopulariopsis*, *Stachyobotris,* and *Trichomonascus* in hay making [20], as well as grain-rich pellets [25]. Dart [25] found that the inoculation of horse pellets, which are rich in cereal grains, with H57 spores at the rate of 10^6^ and 10^7^ cfu/g pellet (as fed), suppressed the development of fungi in the pellets when stored at ambient temperature for 3 months. However, the associated mVOCs were not determined. The aim of this study was to determine the odour profiles associated with microbial contamination in feedlot pellets This study tested the hypothesis that H57 would reduce the production of the mVOC profile that was responsible for the off-odour characteristics of stored, grain-rich (beef feedlot) pellets.

## 2. Materials and Methods

### 2.1. Production of Beef Weaner Pellets

The feedlot pellets were produced at the Ridley Agriproducts Pty Ltd. (Toowoomba, QLD, Australia) commercial feed mill. The un-inoculated pellets were produced prior to the H57 inoculated pellets. The procedure used to prepare the H57 inoculum for this experiment was as described by Schofield [26]. Briefly, the H57 inoculum was prepared in a 100 L fermenter at the University of Queensland and the bacterial spores and vegetative cells were subsequently separated from the supernatant in a Sharples Centrifuge AS26 at 2500 *g* (Sharples Separator Works, West Chester, PA, USA). The pellets were re-suspended in supernatant and mixed with a bentonite carrier which was sieved to 150 μm, frozen at −20 °C and freeze-dried. The material was then ground to a powder in a mortar and pestle and mixed progressively with 200 kg of finely-ground sorghum (<1 mm particle size). The bentonite inoculum contained 5 × 10^10^ spores/g (as well as c.10% vegetative cells) and was then added to the pellet mix at a rate to provide a final concentration of 3.1 × 10^6^ spores/g pellet, as fed. To achieve the final concentration, the freeze-dried H57 bentonite inoculum was first blended with 3 kg of finely ground, extruded wheat in a domestic food mixer, and then added to 30 kg of extruded wheat and mixed in a concrete mixer. This was then added to the 2 tonne batch of pellet ingredients in a commercial paddle mixer. The mixed batch then went through the pelletising and cooling processes before weighing and packing into 20 kg woven plastic bags (LN2, Pacific Bags Australia, Brisbane, Australia). The pellets consisted of sorghum, millrun, full fat soybean, barley, extruded wheat, molasses, limestone, vegetable oil, salt, and a vitamin plus mineral premix (10.0, 66.3, 5.0, 10.0, 2.0, 3.0, 2.0, 1,0, 0.5 and 0.2% dry matter, respectively).

### 2.2. Pellet Storage and Sample Collection

The 20 kg bags that contained the feed pellets to be sampled were stored alongside other bags that would be used in a subsequent calf feeding trial [27]. Approximately 50 bags were stored under ambient environmental conditions in a vermin-proof, closed shed (10.5 × 12.5 m). The bags were elevated from the floor on wooden pallets and separated standing up, each bag at least 10 cm from the next, to allow airflow around the outside of each bag. The remaining bags were stored in a cold room (5.5 × 5.0 m) at approximately 5 °C. The temperature and humidity were recorded within each storage site by a temperature data logger (Easylog USB, Lascar electronics, Wiltshire, UK) (Figure 1). There were 14 pellet treatments that were performed, including:

Pellets stored at ambient temperature (A):
CA0, CA1, CA2 & CA3: Control pellets aged 0, 1, 2 & 3 monthsHA0, HA1, HA2 & HA3: H57 pellets aged 0, 1, 2 & 3 monthsPellets stored at 5 °C (B):CB1, CB2 & CB3: Control pellets aged 1, 2 & 3 monthsHB1, HB2 & HB3: H57 pellets aged 1, 2 & 3 months

Samples of the pellet treatments were taken at the beginning of the storage period and then at 1, 2 and 3 months. The pellet samples were collected by probing 10 bags of each treatment randomly. Each bag was placed upright and then a grain probe (GP-112, Gilson INC, Lewis Center, OH, USA) was inserted diagonally from the top corner to the opposite bottom corner. Approximately 100 g samples were subsequently withdrawn from each bag. The sub-samples were combined and mixed thoroughly to obtain a final sample of 500 g (2 replicates for every pellet treatment at each collection time). Next, the feed samples were ground to a fine powder using an electric coffee grinder (EM0405 MultiGrinder™, Sunbeam, NSW, Australia). Samples of 50 g, for the gas chromatography mass spectroscopy analysis and the plate count, were then taken using a quartering method [28]. Briefly, the fine powder was mixed thoroughly and then poured onto a clean plastic sheet to form an even layer. This layer was marked into quarters and two opposite quarters were kept. These steps were repeated until 50 g total was collected (3 replicates for every pellet treatment at each collection time). The final samples were stored in sealed plastic bags. The pellet samples for the odour profile analysis were stored at −20 °C, while the samples for the plate counts of the H57 populations were held at 4 °C for the subsequent analysis.

### 2.3. Cell Counting Procedure for H57 Spore and Vegetative

The H57 concentrations (spore and vegetative cells) in the pellet samples (3 replicates for every pellet treatment at each collection time) were counted by the viable count method of Harrigan [29]. Briefly, 1.0 g of powdered material was weighed into a sterilised 200 mL beaker, and then 100 mL sterilized chilled water added. The suspension was then mixed for 2 min at 24,000 rpm using a T25 digital Ultra-Turrax IKA homogeniser with a 25 mm dispersing tool (IKA, Staufen, Germany). Three independent 0.1 mL aliquots were taken from the feed suspension and mixed with 0.9 mL sterile water in sterile 1.5 mL Eppendorf tubes. The tubes were labelled as the A, B, and C samples. To count the total numbers of cells, 0.1 mL aliquots (one replicate each from the A, B, and C tubes) were spread on nutrient agar and labelled appropriately. The Eppendorf tubes were then heated for 20 min at 80 °C in a heating block (1572VWR, VWR^TM^, Radnor, PA, USA) to kill vegetative cells, and a repeat 0.1 mL aliquot was then spread onto nutrient agar to count the spores. Cells were grown overnight at 28–30 °C and the colonies on the plate were tabulated. Vegetative cells were determined as the total cells minus spores. The C pellets had no H57. The cell counts of the H57 pellets are shown in Figure 2.

### 2.4. Gas Chromatography Mass Spectroscopy Analysis (GC/MS)

The analysis of VOCs was previously described by Ngo [27]. Briefly, powdered feed pellets (100 mg) were added into 20 mL vials (226-50547-00, Shimadzu, Columbia, MD, USA) and extracted by a Shimadzu GCMS-TQ8040™ (Shimadzu, Kyoto, Japan) using nitrobenzene (100 ppm) as an internal standard (ISTD, 1 ppm; PESTANAL^®^, analytical standard, Sigma-Aldrich, NSW, Australia). The analyses were run on a GC (221-75962-30, Shimadzu, MD, USA), hyphenated to an MS by an SH-Rxi-624Sil MS column, 30 m × 0.25 mm (ID) × 1.4 µm (df). The injection temperature was 200 °C and the analyses were performed with the following programmed temperatures: initially 40 °C, hold for 5 min, a ramp of 20 °C/min to 240 °C, hold for 5 min (total scan time 20 min). The helium gas flow rate was held constant at 1.0 mL/min. Mass spectra were recorded at 0.3 s/mass within a range of 30–350 *m*/*z*. The temperature of the ion source and interface were 200 °C. The VOC compounds were identified by their comparison to a spectral library (National Institute of Standards and Technology—NIST 2014). The data from the VOC analyses (5 replicates for every pellet treatment at each collection time) were described as the peak area of the VOCs detected.

### 2.5. Data Analysis

The VOC data (AU × 10^5^, µm^2^) were analysed by a general linear model (GLM) of the SAS software, Version 9.4 (AS Institute Inc., Cary, NC, USA) [30] using the following model:Y_ijkl_ = μ + Pr_i_ + Tem_j_ + Time_k_ + Pr_i_ Tem_j_ + Pr_i_ Time_k_ + Pr_i_ Tem_j_Time_k_ + ε_ijkl_
where Y is a dependent variable, μ is the overall mean Pr_i_ fixed effect of a probiotic i (i = 1 to 2), Tem_j_ is the fixed effect of the storage temperature (j = 1 to 2), Time_k_ is the fixed effect of the storage time k (k = 1 to 4), Pr_i_ Tem_j_ is the fixed effect of the interaction between the storage temperature and probiotic, Pr_i_ Time_k_ is the interaction between the probiotic and storage time, Pr_i_ Tem_j_Time_k_ is the interaction among the probiotic, storage temperature and storage time, and ε_ijkl_ is the residual error. All interactions were systematically removed from the model when they were non-significant, and a reduced model was used to determine the treatment effects. The solution option with the CLPARM was used to obtain the regression parameter estimates. The effects of the H57 probiotic, storage temperature and storage time were displayed as:Y = b_o_ + b_1_H57 + b_2_Temp + b_3_Time
where Y = dependent variable; b_o_ = intercept of the model; b_1_, b_2_ and b_3_ = coefficient of the H57 probiotic, storage temperature and storage time, respectively. The significance level of the *b*-coefficients was defined at *p* < 0.05 and tendency at *p* < 0.10.

The VOC data were also analysed by a principal component analysis (PCA) using the Unscrambler^®^ X Software, Version 10.5 (Aspen Technology, Inc., Massachusetts, USA) [31], to examine the structural information and possible relationships in the data. Prior to the PCA, the peak areas of each VOC were normalised to a mean of zero and then divided by their total initial standard deviation [32]. The PCA score plot was used to determine the similarities, differences and groupings of the pellet treatments. The PCA bi-plot and correlation loading plots were used to assess the VOCs’ interrelationships and the important VOCs that contributed the most to an observed difference of pellet treatments.

## 3. Results

Forty VOCs were detected in all the samples tested; 24 were of a microbial origin (mVOCs) and 16 were of a potentially non-microbial origin (non-mVOCs) (Table 1). They belonged to at least six different chemical groups including acids, aldehydes, alcohols, ketones, nitrogen compounds, and heterocyclic compounds. Inoculation with H57 significantly reduced the peak areas of five mVOCs (pentanal; 1-pentanol; hexanal; furan, 2-pentyl- and 2,4-decadienal, (E,E)-) and two non-mVOCs (4-cyclopentene-1,3-dione and1,3-hexadiene, 3-ethyl-2-methyl-) (*p* < 0.05), and tended to reduce those of seven mVOCs (propionic acid; furfural; 2-furanmethanol; 1-octen-3-ol; heptanal; 2,4-heptadienal, (E,E)- and nonanal) (*p* < 0.1). In contrast, the addition of H57 increased the peak area of the non-mVOC, butylated hydroxytoluene (*p* < 0.05). A higher storage temperature and longer storage times increased the peak areas of ten mVOCs (pentanal; 1-pentanol; hexanal; heptanal; furan, 2-pentyl-; 1-octen-3-ol; 2,4-heptadienal, (E,E)-; nonanal; (3H)-furanone, 5-ethyldihydro- and 2,4-decadienal, (E,E)-) and three non-mVOCs (1,3-hexadiene, 3-ethyl-2-methyl-; 2-decenal, (E)- and 9,12-octadecadienoic acid (Z,Z)-) (*p* < 0.05).

The first two principal components (PC: 38% and PC2: 22%) explained 60% of the variation between the VOC profiles that was identified in the pellet treatments aged 0, 1, 2, and 3 months (Figure 3). As shown in Figure 3, all of the pellet treatments which were stored at low temperature (cold room) are clearly separated in the positive region of PC1. In contrast, most of the pellet treatments which were stored at high (ambient) temperature (CA1, CA2, CA3, HA2 & HA3), are located in the negative region of PC1. The VOC profile of HA3 resulted in a similar clustering to those of the CA1, CA2, and HA2, while that of CA3 was differentiated from the other pellet treatments and is located in the far northwest quadrant of the PCA score plot.

The associations between each VOC and pellet treatment are revealed in the bi-plot (Figure 4). The separation of CA3 was associated with 19 VOCs. The correlation loading plot (Figure 5) indicated that a significant correlation of VOCs with the pellet treatments was considered when the VOCs were located between the outer (r = 1) and inner (r = 0.5) ellipse. This left 14 VOCs (9 mVOCs and 5 non-mVOCs) that were inferred to be major contributors to the separation of CA3; these are highlighted in bold in Figure 5. The nine mVOCs that were strongly associated with the CA3 pellet treatment were pentanal; hexanal; 2,4-heptadienal, (E,E)-; nonanal; 2,4-decadienal, (E,E)-; 1-octen-3-ol; furan, 2-pentyl-; pyrazine, methyl- and 2(3H)-furanone, 5-ethyldihydro. Additionally, the five non-mVOCs that were related to the CA3 treatment were 9,12-octadecadienoic acid (Z,Z)-; 2-decenal, (E)-; pyrazine, ethyl-; 1-hexyne, 5-methyl- and 1,3-hexadiene, 3-ethyl-2-methyl-.

As presented in the heatmaps (Figure 6), the peak areas of 8 out of the 14 VOCs were found to be slightly higher in the CA3 treatment when compared to that of the other pellet treatments. Of these, the peak areas of 2,4-decadienal, (E,E)-; 1-octen-3-ol and furan, 2-pentyl- were clearly higher in the CA3.

## 4. Discussion

Feedlot pellets are especially susceptible to mould development due to a high starch content and a relatively balanced nutrient profile, and probiotics such as H57 could help to ameliorate this risk. Inoculation with H57 reduced the production of mVOCs that are released by microbial contamination in pellet treatments after 3 months of ageing. This advantage occurred when the pellets were stored at a high (ambient: mean 22 °C) but not at a low (cool room: mean 5 °C) temperature. These results are consistent with the hypothesis that H57 can reduce the rate of mVOC production in grain-rich feed pellets. The VOC profile of the un-inoculated pellets aged 3 months and stored at ambient temperature clearly differentiated from those of the other pellet treatments. The separation of this pellet treatment was mainly due to the contributions of nine mVOCs and five non-mVOCs. Some of these mVOCs have been identified as the causes of off-odours within stored grains that are contaminated by fungi, thereby reducing their freshness, palatability, and safety for livestock.

Nine mVOCs were implicated as those most likely to change in relation to the spoilage of grain-rich pellets. These included aldehydes, alcohols, heterocyclic compounds, and a nitrogenous compound. Although we did not directly identify fungal strains in this current study, these mVOCs may serve as markers for the detection of fungal contamination in grain-rich pellets, as well as indicate the types of fungi that H57 is most able to protect against.

The five mVOCs aldehydes that were identified (pentanal; hexanal; 2, 4-heptadienal, (E, E); nonanal and 2,4-decadienal, (E, E)-) have previously been detected in grains contaminated by fungi. Olsson [33], Williams [34], and Chen [35] reported that the presence of these mVOCs in barley and wheat grains is related to contamination with *Aspergillus flavus*, *A. niger*, *Fusarium graminearum*, *Fusarium culmorum,* and *Penicillium aurantiogriseum*.

The mVOC alcohol that was identified, 1-octen-3-ol, is known to be produced by the enzymatic oxidation and cleavage of linolenic acids, through the action of fungal lipoxygenase and hydroperoxide lyase [36,37]. Together with other eight-carbon mVOCs, 1-octen-3-ol is one of the most abundant mVOCs identified in fungal spoiled grains [36,38]. In fungus-spoiled wheat, 1-octen-3-ol has been positively correlated with the presence of *Aspergillus flavus*, *Aspergillus. ochraceus*, *Aspergillus oryzae*, *Aspergillus parasiticus*, *Aspergillus nidulans*, *Penicillium chrysogenum*, *Penicillium citrinum*, *Penicillium funiculosum*, *Penicllium raistricki*, *Penicillium viridicatum*, *Cephalosporium* spp, and *Fusarium* spp. [39,40].

The two heterocyclic mVOC compounds that were identified (furan, 2-pentyl and 2(3H)-furanone, 5-ethyldihydro-) have previously been linked with fungal growth in wheat and barley grains. An increase in furan, 2-pentyl was linked to the spoilage of wheat grains contaminated by *Aspergillus flavus, A. amstelodami,* and *Penicillium cyclopium* [41], as well as in barley grains contaminated by *Penicillium aurantiogriseum* and *Penicillium verrucosum* [42]. Likewise, 2(3H)-furanone, 5-ethyldihydro- has also been found in durum wheat, bread wheat, and triticale grains that are naturally contaminated by fungi from the genus *Fusarium* [43].

The nitrogenous mVOC compound that was identified, pyrazine, methyl-, has been identified in barley grains contaminated by species of *Aspergillus*, *Penicillium*, *Eurotium,* and *Fusarium* [33]. This mVOC has been proposed to be produced by fungi from the genera *Aspergillus* and *Penicillium* [44] through a dehydration reaction between acetoin and ammonia [36].

Some of the nine mVOCs identified have previously been implicated as “off-odours”. Magan and Evans [36] described the odours produced by nonanal and 2,4-decadienal, (E, E)- in spoiled grains as “musty” and as “fried oil”, respectively. The odour of 1-octen-3-ol has been described as “mouldy” or “raw mushroom” [45], and can vary depending on the presence of different enantiomers [37]. Additionally, Williams [34] found that furan, 2-pentyl was responsible for “spoilage odours” of contaminated wheat grains. Likewise, Olsson [33] indicated that pyrazine, methyl-, together with a number of other mVOCs, caused the development of off-odour characteristics of spoiled barley grains, due to mould growth. Inoculation of pellets with H57 was effective in controlling this odour by inhibiting bacterial and fungal development. This complements the previous observations that H57 inhibits mould development, measured as fungal DNA content, in pelleted feeds stored for three months at 30 °C [25].

In addition to the mVOCs described above, *B. amyloliquefaciens* produces both VOCs and non-volatile compounds, which are known to inhibit the growth of fungi [22,23,46]. These mVOCs include butanal, 3-methyl-; nonanal; 1-pentanol; 2-heptanone; furan, 2-pentyl- and benzaldehyde [22,24,46,47], which were identified in the H57 pellets as well as in the C pellets. Thus, these mVOCs cannot be considered as markers of H57 antimicrobial effects. Schofield [48] found that the H57 genome codes for lipopeptides (non-volatile compounds) such as surfactins, iturins, bacillomycin D, and fengycins. These compounds have antimicrobial activity against a wide range of potential phytopathogenic bacteria and fungi [49].The effect of H57 on mould fungi in pellets could also be mediated through these inhibitory cyclic lipopeptide compounds. A future study could measure their concentrations within pellets by GCMS, as well as with matrix assisted laser desorption/ionization time of flight mass spectrometry (MALDI-TOF MS).

The peak areas of the VOCs that were identified in the pellet treatments were also influenced by the storage temperature alone. The VOC profiles of the pellets were relatively stable at a low (cool room) than at a higher (ambient) temperature, likely because of the effect on bacterial and fungal growth. The influence of temperature on fungal development in stored grains can vary with the fungal species and even within isolates of the same species, with growth of major spoilage fungi species of the genera *Aspergillus*, *Fusarium,* and *Penicillium* occurring across a temperature range of 5 to 35 °C, with optimum temperatures between 20–35 °C [50].

Microbial volatile organic compounds were detected in the pellet treatments at the beginning of the storage period. This suggests that the materials that were mixed to form the pellets were already contaminated by the mould fungi and bacteria that survived the pelleting process, or that contamination occurred after the pellet production, when the pellets were cooled and bagged. The presence of fungi and bacteria in animal feed has been reported by different authors [51,52,53]. Strategies to reduce the original fungal and bacterial commensal load within feed materials, before formulation, should be investigated.

## 5. Conclusions

The suppression of fungal and bacterial contamination within grain-rich pellets by H57 inoculation was shown by the reduction in mVOC production, when pellets were stored at ambient temperature for 3 months. The nine mVOCs that were indicated to be involved deserve further validation as markers of microbial, most likely fungal, spoilage and as markers of the success of the antifungal effects of H57.

## Figures and Tables

**Figure 1 animals-11-03227-f001:**
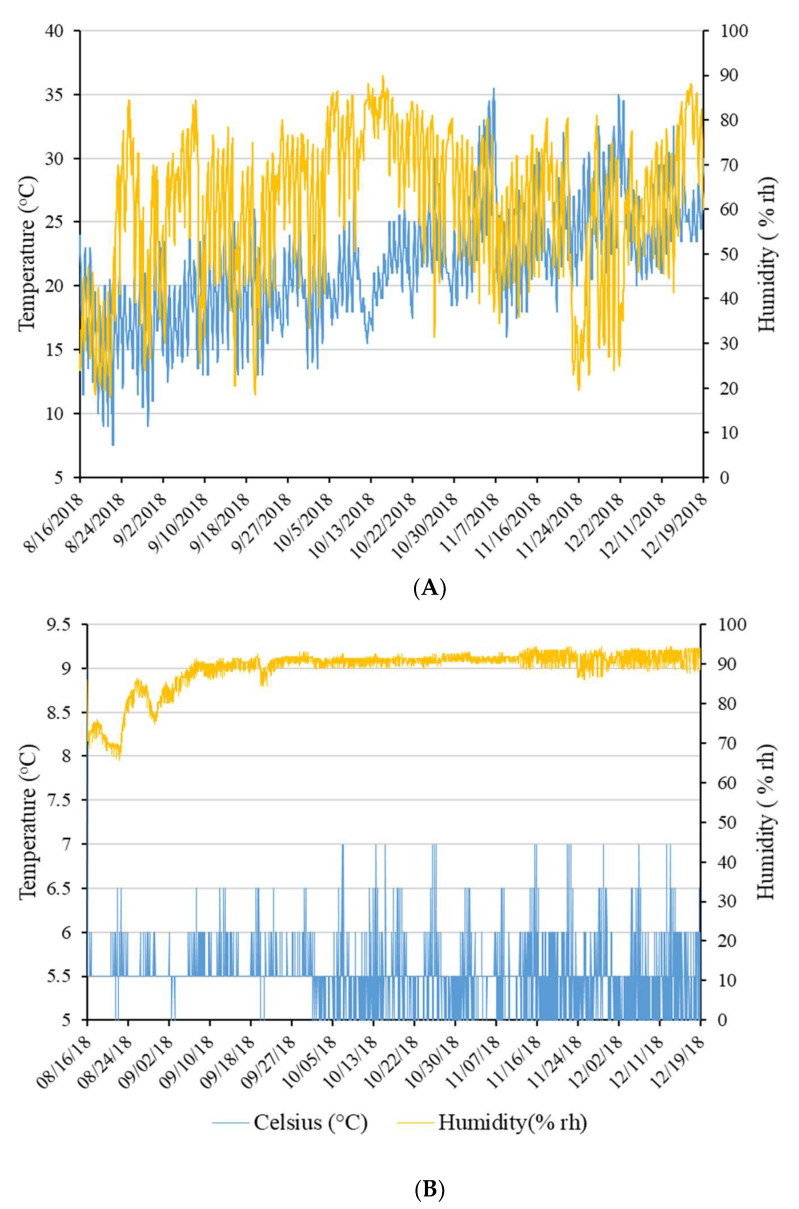
Changes in temperature and relative humidity in the closed shed (**A**) and cool room (**B**) during the experiment period.

**Figure 2 animals-11-03227-f002:**
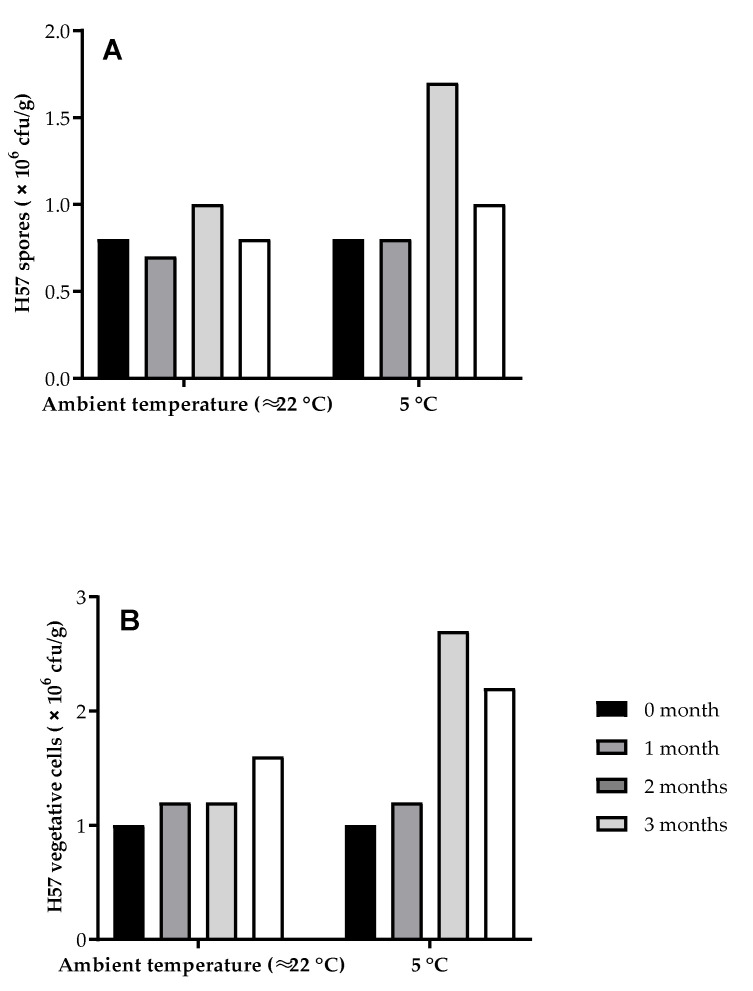
Enumeration of H57 spores (**A**) and vegetative cells (**B**) from the pellet treatments.

**Figure 3 animals-11-03227-f003:**
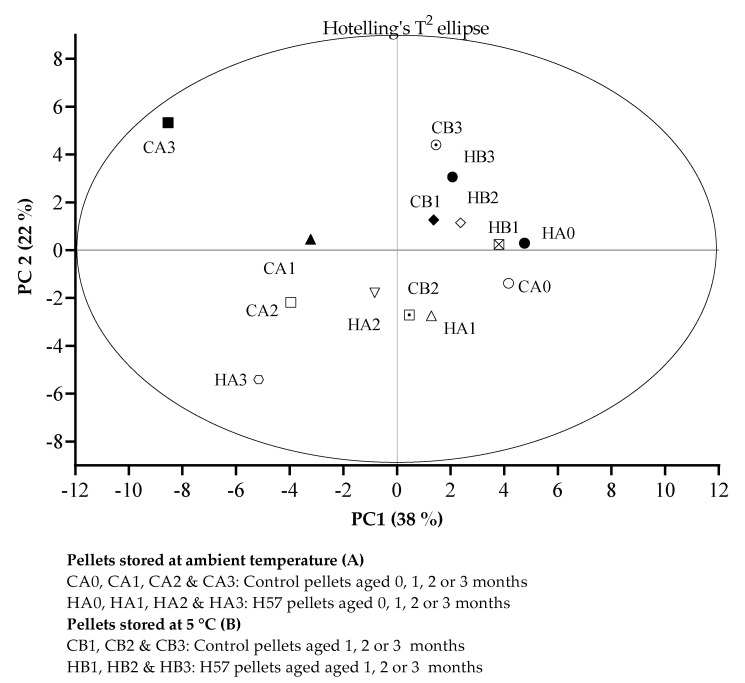
Distributions between the pellets treatments in a score plot of the PCA analysis. Hotelling’s T^2^ ellipse indicated the outlier with critical test value at α = 0.05.

**Figure 4 animals-11-03227-f004:**
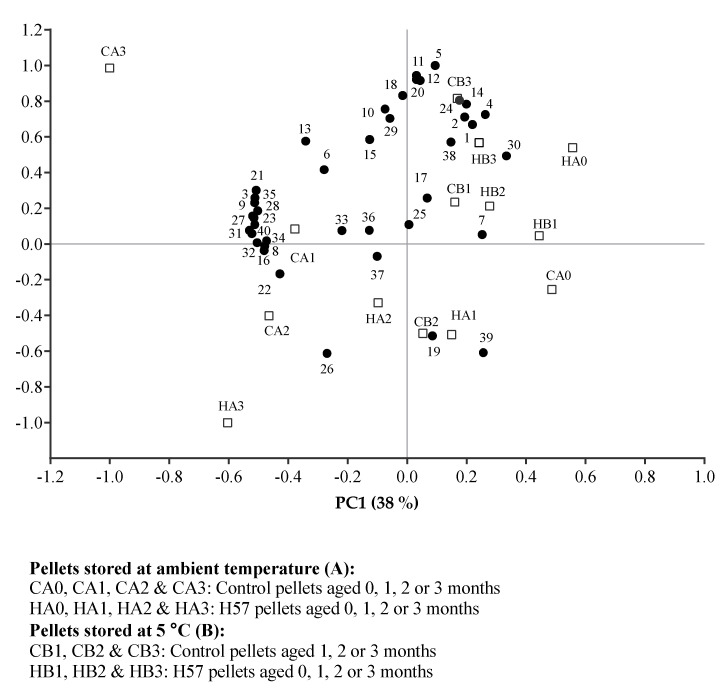
Pattern of correlations between the identified volatile organic compounds and pellet treatments in a bi-plot of the PCA analysis (●VOCs; □ pellet treatments). The numbers correspond to the compound names as in heatmaps of volatile organic compounds.

**Figure 5 animals-11-03227-f005:**
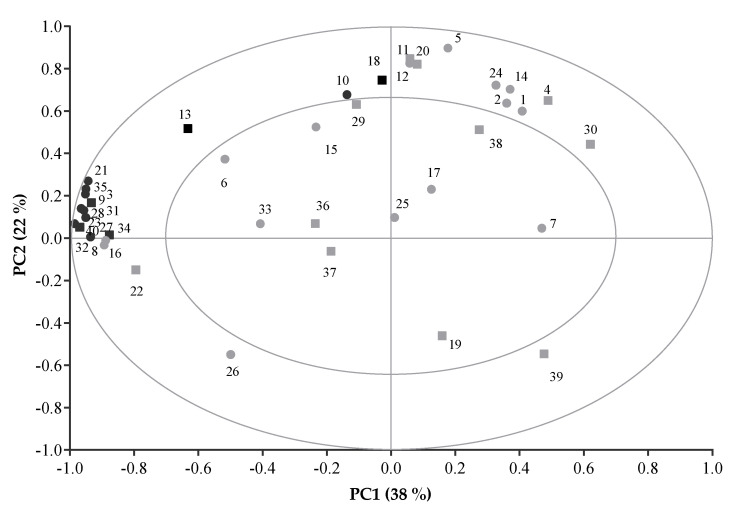
Correlation loading plot of the PCA analysis with 9 important mVOCs and 5 non-mVOCs (in bold) relating to the separation of the CA3 pellet treatment (● & ● mVOCs; ∎ & ∎ non-mVOCs). The numbers correspond to the compound names as in Figure 6. The outer ellipse indicates 100% explained variance. The inner ellipse indicates 50% of explained variance.

**Figure 6 animals-11-03227-f006:**
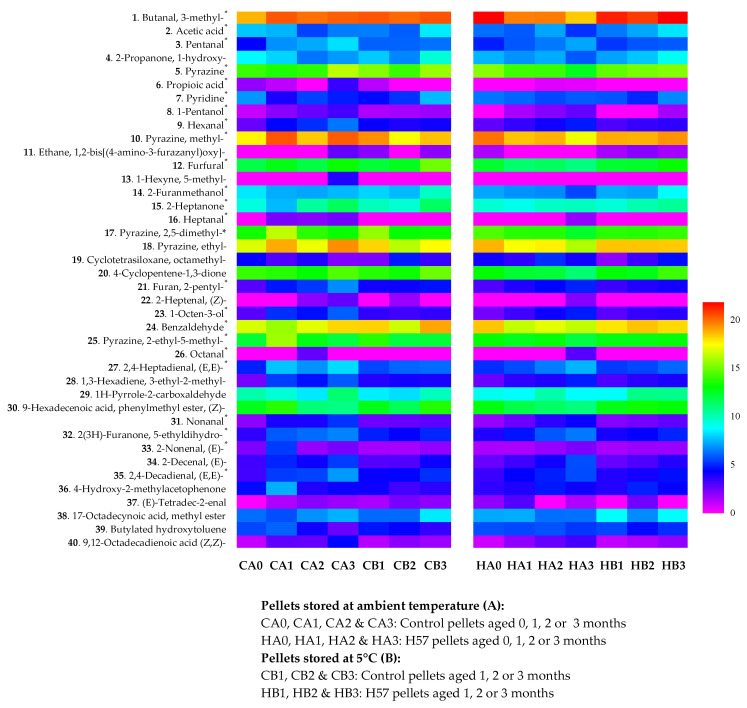
Heatmaps of volatile organic compounds (AU x 10^5^, µm^2^), listed in order of retention time, in the pellet treatments measured by a qualitative analysis of GC/MS. * Microbial volatile organic compounds were identified by details given at http://bioinformatics.charite.de/mvoc/index.php?site=home (accessed on 10 September 2019).

**Table 1 animals-11-03227-t001:** Estimated regression coefficients of peak areas ^1^ of volatile organic compounds in the pellet treatments measured by the GC/MS.

Chemical Classification	Compound Name	Intercept	Coefficient ^2^			*p*		
			H57	Temperature	Time	H57	Temperature	Time
Microbial volatile organic compounds ^3^								
Aldehydes	Butanal, 3-methyl-	21.11	0.48	−1.09	−1.69	0.407	0.102	0.536
Acids	Acetic acid	7.96	−0.22	−0.86	−2.14	0.717	0.212	0.756
Aldehydes	Pentanal	3.82	−0.54	1.06	7.05	0.036	0.002	<0.001
Nitrogen compounds	Pyrazine	15.79	−0.21	−0.96	−1.93	0.719	0.155	0.824
Acids	Propionic acid	0.68	−0.95	0.71	−0.01	0.089	0.218	0.644
Nitrogen compounds	Pyridine	6.95	0.43	−0.51	−4.08	0.391	0.354	0.444
Alcohols	1-Pentanol	−0.39	−0.81	1.30	5.49	0.002	<0.001	<0.001
Aldehydes	Hexanal	2.06	−0.80	1.00	6.13	0.017	0.009	<0.001
Nitrogen compounds	Pyrazine, methyl-	18.56	−0.08	0.22	0.51	0.898	0.757	0.867
Aldehydes	Furfural	13.54	−0.83	−1.01	−0.59	0.083	0.055	0.550
Alcohols	2-Furanmethanol	9.43	−0.88	−1.23	−2.73	0.059	0.021	0.499
Ketones	2-Heptanone	9.93	−0.21	−0.46	0.97	0.624	0.331	0.118
Aldehydes	Heptanal	−1.06	−0.70	1.41	4.23	0.094	0.008	0.024
Nitrogen compounds	Pyrazine, 2,5-dimethyl-	14.15	−0.44	−0.04	−0.07	0.386	0.939	0.425
Heterocyclic compound	Furan, 2-pentyl-	2.32	−0.75	0.98	6.17	0.013	0.005	<0.001
Alcohols	1-Octen-3-ol	1.64	−0.61	1.28	6.11	0.050	<0.001	<0.001
Aldehydes	Benzaldehyde	18.79	0.19	−1.28	−2.53	0.703	0.036	0.677
Nitrogen compounds	Pyrazine, 2-ethyl-5-methyl-	12.47	−0.22	0.28	1.09	0.686	0.638	0.785
Aldehydes	Octanal	−0.93	0.03	0.92	2.75	0.963	0.169	0.257
Aldehydes	2,4-Heptadienal, (E,E)-	4.03	−0.53	1.24	6.13	0.088	0.003	<0.001
Aldehydes	Nonanal	0.71	−0.50	1.40	6.10	0.048	<0.001	<0.001
Others	2(3H)-Furanone, 5-ethyldihydro-	2.49	−0.34	1.36	6.74	0.141	<0.001	<0.001
Aldehydes	2-Nonenal, (E)-	1.30	−0.68	0.85	2.02	0.239	0.181	0.503
Aldehydes	2,4-Decadienal, (E,E)-	2.54	−0.61	1.07	6.40	0.009	<0.001	<0.001
Non-microbial volatile organic compounds								
Ketones	2-Propanone, 1-hydroxy-	9.3	−0.37	−1.06	−3.50	0.487	0.090	0.389
Others	Ethane, 1,2-bis[(4-amino-3-furazanyl)oxy]-	1.73	−0.02	−1.04	−0.74	0.976	0.101	0.736
Others	1-Hexyne, 5-methyl-	−0.36	−0.53	0.62	1.87	0.356	0.321	0.218
Nitrogen compounds	Pyrazine, ethyl-	17.82	−0.13	−0.09	−0.07	0.847	0.903	0.998
Others	Cyclotetrasiloxane, octamethyl-	3.70	0.41	0.25	−1.65	0.409	0.637	0.555
Ketones	4-Cyclopentene-1,3-dione	14.98	−1.25	−0.82	−2.14	0.010	0.077	0.677
Aldehydes	2-Heptenal, (Z)-	−0.51	−0.61	0.82	3.32	0.217	0.135	0.076
Others	1,3-Hexadiene, 3-ethyl-2-methyl-	1.75	−0.65	0.99	6.82	0.042	0.009	<0.001
Heterocyclic compound	1H-Pyrrole-2-carboxaldehyde	9.54	0.19	−0.03	−0.49	0.752	0.967	0.612
Others	9-Hexadecenoic acid, phenylmethyl ester, (Z)-	13.94	−0.02	−1.19	−3.31	0.967	0.053	0.176
Aldehydes	2-Decenal, (E)-	1.78	−0.39	1.34	4.97	0.115	<0.001	<0.001
Others	4-Hydroxy-2-methylacetophenone	3.46	−0.24	0.68	1.94	0.704	0.321	0.736
Aldehydes	(E)-Tetradec-2-enal	0.90	−0.33	0.20	1.84	0.624	0.778	0.727
Others	17-Octadecynoic acid, methyl ester	8.43	0.64	−0.84	−0.27	0.221	0.143	0.524
Alcohols	Butylated hydroxytoluene	4.89	0.97	0.26	−2.29	0.040	0.557	0.071
Acids	9,12-Octadecadienoic acid (Z,Z)-	−0.25	−0.41	1.39	5.96	0.106	<0.001	<0.001

^1^ Peak areas (AU × 10^5^, µm^2^): qualitative analysis by GC/MS; ^2^ H57: H57 probiotic; Temperature: storage temperature; Time: storage time; ^3^ Microbial volatile organic compounds were identified by details given at http://bioinformatics.charite.de/mvoc/index.php?site=home (accessed on 10 September 2019).

## Data Availability

The data presented in this study are available on request from the corresponding author. The data are not publicly available due to privacy.

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
