# Peer review of "Volatile Organic Compound Profiles Associated with Microbial Development in Feedlot Pellets Inoculated with Bacillus amyloliquefaciens H57 Probiotic"

_animals, 2021, doi:10.3390/ani11113227_

Round 1

Reviewer 1 Report

The aim of these studies was to test the influence of probiotic Bacillus amyloliquefaciens H57 on the odour profile of stored feedlot pellets. This purpose was clearly described and justified in the Introduction to the manuscript.

The manuscript is extremely interesting and well-written, however, in my opinion, it requires minor corrections and supplementing the data before allowing it to be published:

  • Line 107: please convert "17000 rpm" to value "x g".
  • Line 108: please check if the unit given at "150μ" is correct.
  • Line 111: are you sure that the bentonite inoculum contained only spores? There were no vegetative cells there?
  • Lines 118-121: please check that the given value is correct for each component.
  • Lines 121, 165, 190, and Figure 2: please add information about the number of repetitions of the experiment or analysis.
  • Line 221 and further fragments of the text: why do you write on "the peak areas" when internal standard was used? Why don't you show the results of the quantitative analysis?
  • Lines 225-226: how should this phenomenon be explained?
  • Table 1: please add the results of the qualitative or quantitative analysis of the GCMS. I suggest to organize the order of substances additionally according to chemical classification.
  • Lines 237-244: are the Principal Components values obtained too low? How should this analysis be interpreted in practice?
  • Figure 6: what is the unit of the coloured scale in the Figure 6?
  • Lines 277-279: have you tested the microbiological purity of these pellets? In my opinion, such a study is necessary. Is this sentence not exaggerated?
  • Lines 282-288: if you have only marked the peak areas, that is, have made a qualitative analysis, you cannot write such conclusions. Only the quantitative analysis allows for the formulation of such conclusions. I do not find the results of the quantitative analysis in this manuscript.

Author Response

Dear Reviewer,

Thank you for your letter and constructive comments concerning our manuscript entitled “Volatile organic compound profiles associated with microbial development in feedlot pellets inoculated with Bacillus amyloliquefaciens H57 probiotic”. We have studied your comments carefully and made major corrections which we hope meet with your approval. We answer your questions or comments in detail in the following attached file.

Kind regards,

TT Ngo

Reviewer 2 Report

Interesting work, prepared in a clear, readable manner. Includes well described methodology and extended discussion.

I have a comment on the selection of literature - there are many works older than 15 years.

Minor comments below.

L35: „40” instead of „forty”

L97-98: it is worth detailing the purpose of the work, from a new paragraph.

L118-121: this was measured in some way? These parameters?

Figure 1B: the Y-axis does not have to start at 0, I think it would be clearer if it started at e.g. 4.0. Same as in Figure 1A.

L134-137: I suggest to distinguish abbreviations/sample names so that they can be easily referred to in the text and are readable. Here they get a bit lost in the large amount of text.

L137-141: how many samples were finally obtained and were the analyses performed in replicate?

Figure 2: ambient temperature – which one? Worth stating with a number.

L194: the calculation model is the authors' design or comes from some publication?

L218: 40” instead of „forty”

Figure 4 and 5: in the caption there is a reference to figure 6, it is a bit incorrect to refer to a future, next figure and not the previous one. Maybe swap the order?

Figure 6: please explain what the scale is.

L351-354: Aren't there any papers in which researchers came to similar conclusions?

References: As many as 20 papers are from before 2005, which is quite a lot.

Author Response

(The authors gave the same response as above.)

Round 2

Reviewer 1 Report

I see that the authors tried to improve the manuscript according to the reviewers' suggestions, taking into account some of them and others rejecting. I do not have more comments to this text.

Author Response

Dear Reviewer,

We are grateful to you for your time and constructive comments on our manuscript. We look forward to the outcome of the assessment.

Yours sincerely, 
On behalf of the co-authors 
TT Ngo